# The relation between circulating levels of vitamin D and parathyroid hormone in children and adolescents with overweight or obesity: Quest for a threshold

Golaleh Asghari[1], Emad Yuzbashian[1], Carol L. Wagner[2], Maryam Mahdavi[3], Roya Shamsi[1], Farhad Hosseinpanah[3]*, Parvin Mirmiran[1]*

1 Nutrition and Endocrine Research Center, Research Institute for Endocrine Sciences, Shahid Beheshti University of Medical Sciences, Tehran, Iran, 2 Department of Pediatrics, Division of Neonatology, Shawn Jenkins Children's Hospital, Charleston, SC, United States of America, 3 Obesity Research Center, Research Institute for Endocrine Sciences, Shahid Beheshti University of Medical Sciences, Tehran, Iran

* mirmiran@endocrine.ac.ir (PM); fhospanah@endocrine.ac.ir (FH)

**Data Availability Statement:** The data set is the property of Research Institute for Endocrine Sciences (RIES) and is made available upon

## Abstract

The level of serum 25-hydroxyvitamin D (25(OH)D) at which intact parathyroid hormone (iPTH) is maximally suppressed (suppression point) and below which PTH begins to rise (inflection point) has been used to define optimum 25(OH)D concentration. We aimed to study the association of circulating iPTH with 25(OH)D concentrations and to determine a 25(OH)D threshold associated with a significant iPTH suppression. This cross-sectional study was conducted on 198 boys and 180 girls, aged 6–13 years with BMI ≥ 1SD (WHO criteria) recruited from primary schools. Adjusted iPTH for BMI z-score, pubertal status, and dietary calcium was used. Nonlinear regression was used to model the relationship between 25(OH)D and iPTH and identify a suppression point for 25(OH)D at which iPTH reached a plateau. Piecewise regression analysis with a single knot for all possible values of 25(OH)D were fitted. Furthermore, 95% confidence intervals (95%CI) for those point had been calculated. The mean age (SD) of girls and boys was 9.1 (1.6) and 9.4 (1.7) years, respectively. Median 25(OH)D and iPTH were 13.8 ng/mL and 33.9 pg/mL in boys and 9.9 ng/mL and 47.8 ng/mL in girls, respectively. The equation in girls was: log-iPTH = 3.598+0.868 exp [(-0.190×25(OH)D. The point for near maximal suppression of iPTH by 25(OH)D for girls occurred at a 25(OH)D concentration of 20 ng/mL (95% CI: 7.1 to 32.2). No point of maximal suppression was found for boys. We also found a 25(OH)D threshold of 10 ng/mL (95% CI: 4.6 to 22.5) for girls (f: 9.8) by linear piecewise regression modeling of adjusted iPTH. No significant inflection point for boys was observed. In overweight/obese girls, when the concentration of 25(OH)D was higher than 20 ng/mL, an iPTH mean plateau level is reached, and when its concentrations approach 10 ng/mL, the slope of iPTH concentration has been accelerated.

approval of the research proposal by the research council and the ethics committee. The RIES ethics committee must issue an approval in case of a request for access to the de-identified dataset. Data request may be sent to the head of the RIES Ethics Committee, Dr. Azita Zadeh-Vakili, at email: azitavakili@endocrine.ac.ir.

**Funding:** This study was supported by the Shahid Beheshti University of Medical Sciences, Tehran, Iran (Grant no. 10429-4). The funder had no role in study design, data collection and analysis, decision to publish, or preparation of the manuscript. There was no additional external funding received for this study.

**Competing interests:** The authors have declared that no competing interests exist.

**Abbreviations:** iPHT, Intact parathyroid hormone; 25(OH)D, 25-hydroxyvitamin D; BMI, body mass index; EAR, estimated average requirement.

## Introduction

Although vitamin D is known as a micronutrient, it also functions as a hormone. Effects of vitamin D on the regulation of bone metabolism and calcium homeostasis are well-established.

In addition to calcemic function, recent studies indicate that vitamin D has a crucial role in normal growth and puberty, regulation of immune response, cancer prevention, and controlling insulin metabolism [1]. These widespread functions highlight the importance of maintaining optimal vitamin D levels in adults as well as children [2]. It is evident that serum 25-hydroxyvitamin D (25(OH)D) concentration, given its longer half-life of 2–3 weeks, has been considered as the best indicator of overall vitamin D status [3].

The inverse non-linear association between 25(OH)D concentration and parathyroid hormone (PTH) is commonly considered to define the appropriate cut point for defining adequate vitamin D status in children and adults [4–6]. Studies indicated that there is two thresholds of 25(OH)D rather than a single inflection point in the 25(OH)D-PTH-association curve where: 1) the 25(OH)D concentration is high enough to suppress the PTH concentration and 2) 25(OH)D has dropped enough to reciprocally increase PTH. In fact, the former point is a threshold point for the 25(OH)D concentration at which serum PTH concentrations decrease and reach a plateau, and the latter point is a spot at which the intensity of PTH concentration in response to increasing 25(OH)D concentration dramatically changes and slowly reaching a maximal suppression point. Those points at which 25(OH)D maximally suppresses or rapidly raises PTH could be targeted markers for definition of vitamin D deficiency. There is a wide range for achieving PTH suppression with the inflection point of serum 25(OH)D from 8 to 44 ng/ml in adults [7] and 11 to 43 ng/mL for adolescents [8, 9].

Significant controversy exists regarding optimal vitamin D status in children and adolescents, which is complicated by certain factors such as excess weight. Alteration of the vitamin D endocrine system in obesity has been reported [10]. Excess body weight or fat accumulation in both adults and children are associated with lower 25(OH)D concentrations and higher PTH concentrations [11–14]. The 25(OH)D-PTH association may not be explained by the same mechanism in normal-weight individuals. It is not known whether the 25(OH)D-PTH association is affected by obesity; however, there may be a different set-point for the 25(OH)D-PTH relationship in the obese pediatrics. Therefore, determining the threshold for 25(OH)D in children and adolescents with excess weight is more complex, and defining cut points in this population seems crucial. To the best of our knowledge, few studies investigated the association between 25(OH)D and PTH in boys and girls with excess weight. Therefore, the aim of the present study was to determine the maximal suppression point for serum iPTH in obese/overweight adolescent girls and boys. We also aimed to define a point at which the intensity of iPTH changes in response to the 25(OH)D concentration.

## Method and materials

### Study design

The present cross-sectional study was conducted in Iran, Tehran located at 51° 24′E, 35° 42′N from June 2016 to March 2017. Children and adolescents aged 6 to 13 years, with an age- and sex-specific body mass index (BMI) Z-scores ≥ 1 (according to criteria established by the World Health Organization), were recruited from primary schools located in three districts of Tehran. None of the adolescents had diabetes or other known medical illnesses such as liver or kidney diseases, associated with vitamin D metabolism (based on physician examination and medical records review), or used medication or supplements that might affect calciotropic

hormones, or made intentional changes of dietary intake, or physical activity. An alphabet list of all eligible students was prepared and then a simple random sampling was generated. Finally, 180 girls and 198 boys meeting selection criteria were enrolled in the study. All children and adolescents and their guardians were invited to the Research Institute for Endocrine Sciences (RIES). The participants answered all questionnaires including socio-economic and health related issues, physical activity, and dietary intake. Height and weight were measured, and BMI was calculated. Stage of puberty was determined and a fasting blood sample was gathered.

Parents gave written informed consent, and all children provided assent to participate. Ethics approval was obtained from the ethics committee of the Research Institute for Endocrine Sciences (RIES) of the Shahid Beheshti University of Medical Sciences (NO: IR.SBMU.ENDO-CRINE.REC.1395.373).

## Measurements

Laboratory evaluations were performed on 5 mL venous blood samples drawn after overnight fasting (about 10 to 12 h) in the morning. All blood analyses were carried out at the RIES research laboratory. Intact parathyroid hormone (iPTH) and 25(OH)D concentrations were determined by the electrochemiluminescence immunoassay (ECLIA) method, using Roche Diagnostics kits and the Roche/Hitachi Cobas e-411 analyzer (Roche Diagnostics, GmbH, Mannheim, Germany). All intra- and inter-assay CVs were <2.6% for iPTH and <7.5% for 25 (OH)D concentrations. Calcium and phosphorus were measured using the photometric method by arsenazo III and the UV photometric method respectively. Alkaline phosphatase was measured using kinetic photometric, standardized by DGKC. All calcium, phosphorus, and alkaline phosphatase were measured by Pictus 700 Clinical Chemistry Analyzer, Diatron MI PIc (Budapest, Hungry) Parsazmoon kits (Tehran-Iran).

Weight was measured with participants minimally clothed and no shoes, using BIA (JIAI 359 Manufacture of Zhan Korea Co.). Height was measured with participants standing in front of a standard board measurement and arms at their sides. Body mass index was calculated as weight (kg) divided by height ($m^2$). The percentage error in weight and height was 100 g and 0.5 cm, respectively; values obtained in this calculation were converted to Z-scores, according to the curves of growth of the World Health Organization (WHO). Children who presented with BMI Z-scores, based on age- and sex-specific cutoff values, $\geq$ 1 SD were defined as overweight and those $\geq$2 SD were defined as obese [15].

Subjects' physical activity was assessed using the Modifiable Activity Questionnaire (MAQ) [16], confirmed by their parents, especially their mothers, including items on the frequency, duration, and type of exercise performed during the subjects' leisure time. An endocrinologist examined boys and girls to determine the stage of puberty. For girls, based on the criteria for breast assessment and for boys based on their genital standard in 5 stages, both groups were divided accordingly [17, 18]. Exposure to sunlight was estimated using a questionnaire on daily duration of exposure to outdoor sunlight. Low sun-exposure was considered as exposing to sunlight<15min/d.

Dietary intake was collected using three 24-hour dietary recalls (2 weekdays and 1 weekend day) during face-to-face interviews, by a specifically trained dietary interviewer. Participants were asked to report their consumption of food during the last day, and mothers were asked about the type and quantity of foods and ingredients when children were unable to recall. Portion sizes or household measures for each food item were converted to grams. All consumed food items were analyzed for their energy and nutrient content using a nutrient database (Nutritionist 4), which was modified according to the Iranian Food Composition Table.

## Statistical analysis

Characteristics of participants were expressed as percentages for categorical variables, mean (SD) for normally-distributed variables, or median (interquartile range) for non-normally distributed variables. Differences in characteristics between girls and boys were tested using the Student's t-test and Chi-square test as appropriate.

Serum concentration of iPTH was log transformed to approximate a normal distribution. In order to adjust for potential confounders (BMI z-score, pubertal status, and dietary calcium), an adjusted sex-specific variable (because of considerable difference between boys and girls in PTH concentration [19]) was generated for iPTH using its mean value plus the residuals obtained from regressing the iPTH level based on aforementioned confounding factors. The adjusted iPTH concentrations were used for the following analyses.

The shape of the associations between 25(OH)D values and adjusted log-iPTH was explored by restricted cubic splines instead of using arbitrary predetermined cut-points. Restricted cubic splines were used with 3 knots defined at the 25th, 50th, and 75th centiles of 25(OH)D value; these showed a nonlinear relationship between iPTH and 25(OH)D concentration. Nonlinear regression was used to model the relationship between 25(OH)D and adjusted log-iPTH and to identify a suppression point in 25(OH)D where the iPTH reached a plateau and was maximally suppressed.

Piecewise linear regression was used to assess a single threshold point where the intensity of adjusted log-iPTH concentration response to increasing trend of 25(OH)D concentration changed. This threshold was considered in values greater than the driven plateau point obtained from nonlinear regression. A series of piecewise linear regression analyses with a single knot for all possible values of 25(OH)D were fitted. The optimal threshold value was chosen based on adjusted $R^2$, the F statistic, model SE, and the t value and associated P value for the threshold variable. In addition, bootstrap resampling (n = 5000) has been used to determine the %95 confidence interval around the point estimate of serum 25(OH)D for point of plateau and rapidly rise of PTH.

To investigate differences in characteristics between girls with a serum 25(OH)D concentrations less than or greater than the given thresholds based on the point of plateau and inflection of serum adjusted log-iPTH, we used one-way ANOVA with Bonfferoni for continuous and Chi-square test for categorical variables. Furthermore, since there was no point of plateau or inflection in boys, mean adjusted log-iPTH concentration was used to define the points [6], and differences in characteristics of boys below and above the 25(OH)D point were assessed by Student's t-test and chi-square test in the correct position.

A number of sensitivity analyses were conducted using nonlinear regression. These included repeating analyses: (1) excluding participants who consumed above age-specific estimated average requirement (EAR) of calcium (EAR; aged 4 to 8 years = 800 mg, aged 9 to 13 years = 1100 mg) to address whether dietary calcium may affect the association between serum 25(OH)D and PTH; (2) excluding participants who consumed above age-specific EAR of magnesium (EAR; aged 4 to 8 years = 110 mg, aged 9 to 13 years = 200 mg); (3) Excluding participants with overweight (BMI Z-score: 1 to 2), remaining participants with obesity (BMI Z-score $\geq$ 2) to account for the influence of obesity on serum iPTH concentrations.

All statistical analyses were performed using STATA version 12 (STATA, College Station, TX) and IBM SPSS (version 20, Chicago, IL, USA); the significance level was set at P<0·05 (two-tailed).

## Results

The study included 198 boys and 180 girls with a mean age of 9.4 and 9.2 years, respectively. Unadjusted median (25–75 IQR) of 25(OH)D and iPTH concentrations were 13.8 (10.0–19.6)

ng/mL and 33.1 (22.3–48.6) pg/mL, respectively, among boys and 9.9 (6.4–15.8) ng/ml and 47.8 (32.8–76.1) pg/mL, respectively, among girls. Clinical and biochemical sex-based characteristics of participants are shown in Table 1. The prevalence of obesity and BMI z-score among boys was higher than girls ($p < 0.001$). There were higher serum concentrations of PTH and lower 25(OH)D in girls, compared to boys ($p = 0.036$ and $p < 0.001$, respectively). Regarding dietary intakes, the boys consumed more energy ($p < 0.001$), calcium ($p = 0.001$), vitamin D ($p = 0.012$), magnesium($p = 0.002$), and phosphorous ($p < 0.001$) in comparison to girls.

Based on nonlinear regression models, the relationship between 25(OH)D and iPTH was exponential. The estimated fixed effects (and standard error) were: a = 3.598 (0.110), b = 0.868 (0.465), c = 0.190 (0.141) in girls, and a = 3.092 (4.495), b = 0.553 (4.338), c = 0.014 (0.151) in boys. Final equations were as follows:

In girls (Fig 1A): iPTH (pg/ ml) = $3.598 + 0.868 \times \exp^{(-0.190 \times 25(OH)D)}$ and a plateau in iPTH level at 44 pg/mL was observed at a serum 25(OH)D concentration of approximately 20 ng/mL (95% CI: 7.1 to 32.2).

In boys (Fig 1B): iPTH (pg/ ml) = $3.092 + 0.553 \times \exp^{(-0.014 \times 25(OH)D)}$ and there was no plateau in serum iPTH as 25(OH)D concentration increased.

Since there was no plateau for iPTH in boys, depending on a mean serum log-iPTH concentration of 3.53 pg/mL, subjects were divided into two groups [6]. Based on the exponential equation given above, it can be calculated that in boys, a mean serum log-iPTH concentration of 3.53 pg/mL obtained at serum 25(OH)D concentrations of 15 ng/mL.

Piecewise linear regression modeling of log PTH for 25(OH)D showed that in the slope of iPTH-25(OH)D, iPTH began to rapidly rise at 10 ng/mL (95% CI: 4.6 to 22.5) of 25(OH)D level among girls (f: 9.8). However, in boys there was no point in which the slope of the line

**Table 1. Baseline characteristics of participants according to sex.**

|  | Girls | Boys | P-value |
|---|---|---|---|
| Age (years) | 9.1 (1.6) | 9.4 (1.7) | 0.181 |
| Pre-pubertal (%) | 15.2 | 21.2 | 0.131 |
| Physical activity (MET/h/w) | 4.4 (1.0–13.9) | 15.3 (4.9–37.7) | <0.001 |
| Obesity (%) | 56.2 | 80.8 | <0.001 |
| Body mass index z-score | 2.2 (1.9–2.7) | 2.7 (2.1–3.1) | <0.001 |
| Sun-exposure more than 15 min/d (%) | 55.6 | 72.5 | <0.001 |
| **Biochemical assessments** |  |  |  |
| 25-hydroxy vitamin D (ng/mL) | 9.9 (6.4–15.8) | 13.8 (10.0–19.6) | <0.001 |
| Parathyroid hormone (pg/mL) | 47.8 (32.8–76.1) | 33.1 (22.3–48.6) | 0.019 |
| Calcium (mg/dL) | 10.0 (0.7) | 10.2 (0.8) | 0.045 |
| Phosphorous (mg/dL) | 5.0 (0.8) | 4.9 (0.8) | 0.204 |
| Alkaline Phosphatase (IU/L) | 697 (198) | 716 (220) | 0.368 |
| **Dietary intakes** |  |  |  |
| Energy (kcal) | 1620 (498) | 1926 (623) | <0.001 |
| Calcium (mg) | 607 (285) | 724 (390) | 0.001 |
| Vitamin D (IU) | 45.0 (48.8) | 61.9 (74.4) | 0.012 |
| Magnesium (mg) | 182 (80) | 213 (104) | 0.002 |
| Phosphorous (mg) | 807 (283) | 1002 (406) | <0.001 |

Data are represented as mean±SD or median (IQ 25–75) for continues variable and percent for categorical variables.

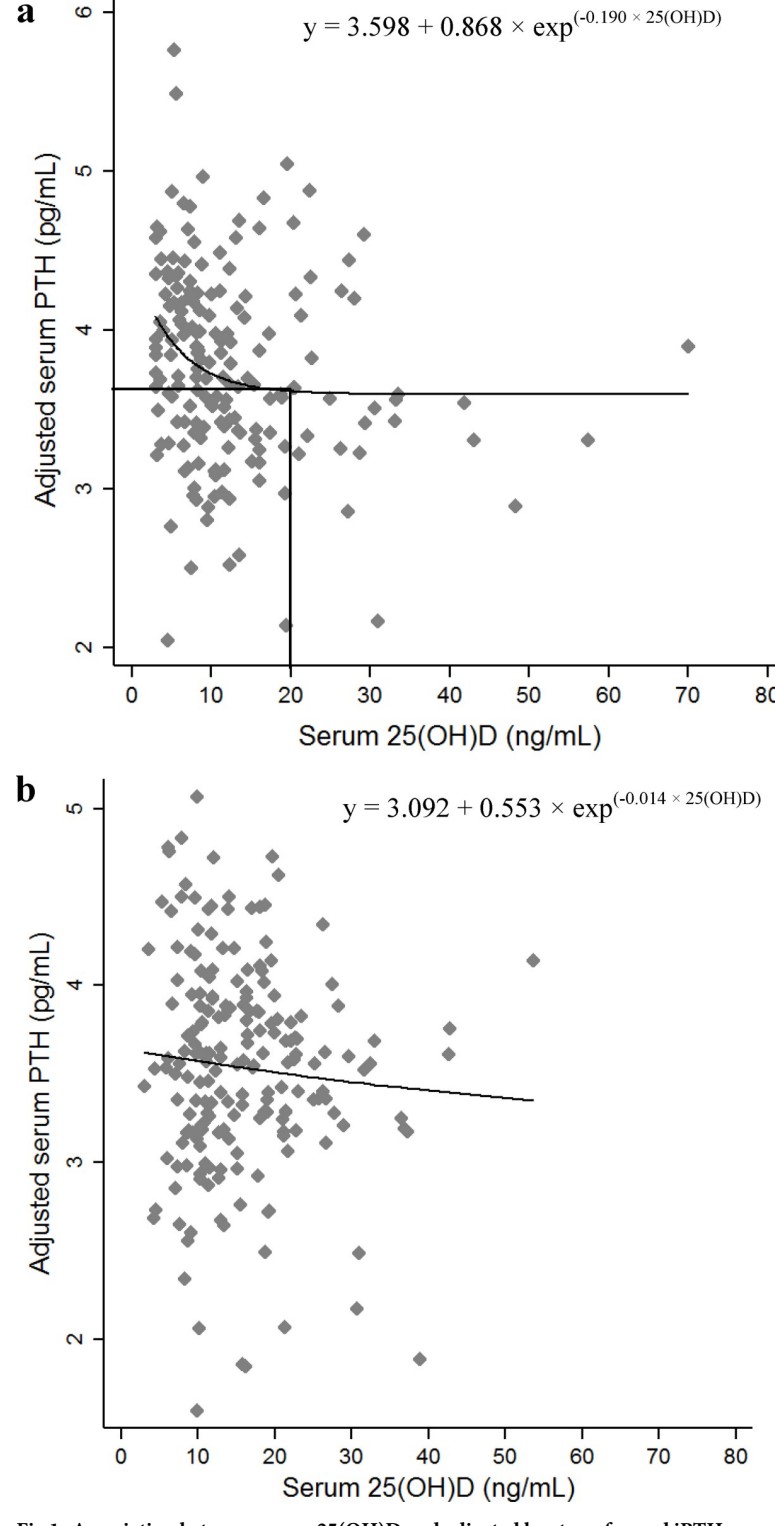

**Fig 1. Association between serum 25(OH)D and adjusted log-transformed iPTH concentrations using nonlinear regression analysis.** (a) females; (b) males.

**Table 2. Characteristics of study participants based on threshold of 25(OH)D concentration (ng/mL).**

| | Girls | | | Boys | |
|---|---|---|---|---|---|
| | <10 (n = 90) | 10 to 20 (n = 58) | ≥20 (n = 29) | 15< (n = 106) | ≥15 (n = 91) |
| Age (years) | 9.7 (1.6)[ab] | 8.2 (1.4)[b] | 9.1 (1.3) | 9.0 (1.6)[c] | 9.8 (1.8) |
| Pre-pubertal (%) | 11.1 | 20.7 | 17.2 | 12.3 [c] | 30.8 |
| Physical Activity (MET/h/w) | 4.3 (1.4–13.4) | 4.6 (0.9–14.5) | 4.2 (0.4–15.2) | 14.6 (4.6–34.3) | 18.0 (5.0–47.3) |
| Obesity (%) | 43.3 | 44.8 | 44.8 | 82.1 | 79.1 |
| Body mass index z-score | 2.3 (0.5) | 2.4 (0.5) | 2.3 (0.5) | 2.8(0.9) | 2.7(0.7) |
| **Biochemical** | | | | | |
| 25-hydroxy vitamin D (ng/mL) | 6.5 (4.7–8.1)[ab] | 13.0 (11.5–16.0)[b] | 27.3 (22.2–33.1) | 10.2 (8.4–11.9) [c] | 20.2 (17.7–25.7) |
| Parathyroid hormone (pg/mL) | 58.4 (38.9–84.9)[a] | 37.2 (26.2–53.9) | 39.4 (32.2–80.4) | 33.0 (21.5–52.7) | 33.1 (22.5–45.6) |
| Calcium (mg/dl) | 9.8 (0.6)[b] | 9.9 (0.9) | 10.2 (0.7) | 10.1 (0.7) | 10.2 (0.8) |
| Phosphorous (mg/dL) | 5.0 (0.5)[b] | 5.0 (0.6) | 5.3 (1.4) | 4.8 (0.6) [c] | 5.1 (0.8) |
| Alkaline Phosphatase (IU/L) | 745 (214)[ab] | 640 (171) | 662 (163) | 714 (234) | 719 (205) |
| **Dietary intakes** | | | | | |
| Energy (kcal) | 1615 (499) | 1555 (493) | 1755 (505) | 1936 (680) | 1913 (557) |
| Calcium (mg) | 586 (257) | 606 (258) | 697 (370) | 756 (389) | 686 (375) |
| Vitamin D (IU) | 36.7 (43.9) | 52.4(52.2) | 54.5 (52.7) | 72.6 (81.3) [c] | 49.6 (63.7) |
| Magnesium (mg) | 175(71)[b] | 178 (69) | 211 (116) | 230 (115) [c] | 193 (87) |
| Phosphorous (mg) | 790 (281) | 827 (288) | 824 (288) | 1045 (402) | 957 (408) |

Data are represented as mean±SD or median (IQ 25–75) for continues variable and percent for categorical variables.

[a] Significant difference compared to 25(OH)D between 10 to 20 ng/mL in girls.

[b] Significant differences compared to 25(OH)D ≥20 ng/mL in girls.

[c] Significant differences compared to 25(OH)D ≥15 ng/mL in boys.

considerably changed. Participants were categorized based on sex-specific cut points of 25 (OH)D concentrations (Table 2). Girls with serum 25(OH)D concentrations <10 ng/mL (point of the threshold) had significantly higher iPTH (p = 0.004) and alkaline phosphatase (p = 0.002) concentrations than girls with serum 25(OH)D concentrations ≥20 ng/mL (point of the plateau). More boys with 25(OH)D concentration ≥15 ng/mL were pre-pubertal and had higher serum phosphorus concentrations than those with serum 25(OH)D concentrations <15 ng/mL. Furthermore, boys with higher concentrations of 25(OH)D consumed less vitamin D and magnesium.

In order to evaluate robustness of findings, sensitivity analyses of participants were also performed (Fig 2). After excluding participants who consumed above the EAR of dietary calcium, the point of the plateau was decreased in girls and reached 17 ng/mL of 25(OH)D. Our findings also showed that the point of plateau among participants who were obese or consumed less than the EAR of magnesium was not substantially different from findings for the whole population. Furthermore, there was also no plateau for boys.

## Discussion

The current study was conducted to demonstrate the inflection point in the relationship between serum 25(OH)D and iPTH in children and adolescents with age- and sex-specific BMI Z-scores ≥ 1. Findings in girls indicated that 20 ng/mL (95% CI: 7.1 to 32.2) of 25(OH)D was the inflection point where the iPTH levels were maximally suppressed and reached a plateau. In contrast, in boys a point of inflection was not observed. Furthermore, by decreasing 25 (OH)D to 10 ng/mL (95%CI: 4.6 to 22.5), the slope of 25(OH)D-iPTH relation changed and started to rise rapidly among girls, but not among boys.

When calcium absorption in the gut does not work efficiently because of the hypovitaminosis D, PTH concentration enhances this process. High concentrations of PTH have adverse effects on the skeleton as well as cardio-metabolic factors [20–24]. The effect of the reciprocal relationship between vitamin D and PTH on bone hemostasis has been considered to define a threshold for adequacy of vitamin D status—the serum 25(OH)D concentration at which serum PTH is suppressed maximally [25]. The Pediatric Endocrine Society defined 25(OH)D deficiency and insufficiency < 15 and < 20 ng/ml, respectively [26]; whereas, the Endocrine Society gives a different definition for 25(OH)D deficiency and insufficiency; i.e., < 20 ng/mL and between 21 to 29 ng/mL, respectively [3]. Another cut point is recommended by Institute of Medicine (IOM); i.e a 25(OH)D concentration < 20 ng/mL is considered inadequate and > 20 ng/mL is considered as an adequate value [27]. It should be noted that due to alterations in vitamin D homeostasis in individuals with excess weight [28], defining sufficiency and insufficiency status of vitamin D in these children should be considered. Children with obesity may be defended against consequences of low 25(OH)D concentration by keeping lower levels of PTH secretion; therefore, they may need a lower 25(OH)D concentration for the maintenance of bone metabolism and calcium homeostasis [29]. In this regard, data on adolescents with excess weight is limited. It was shown that optimal 25(OH)D concentration at which iPTH reached maximal suppression (plateau point) was lower in children with excess weight (12.4 ng/ml) in comparison to that for normal weight children (17.0 ng/ml) [29]. The current study extracted a threshold of 25(OH)D in a sample of adolescents with overweight or obesity and provided further data in this context. The most distinguishing difference is that the median of serum 25(OH)D and iPTH concentrations in their study were 25.5 ng/mL and 30.0 pg/mL, respectively which is higher than what was observed in our study: 9.9 ng/mL and 47.8 pg/mL in girls and 13.8 ng/mL and 33.1 pg/mL in boys, respectively. Thus, these differences in 25(OH)D and iPTH concentration may explain the distinct point of plateau between the studies.

In this regard, to define optimal levels of 25(OH)D at which PTH plateaus or rapidly raises, some studies found no optimal threshold value [6, 30], whereas others report a wide range of cut points, 14 to 43 ng/mL [6, 8, 9, 29, 31–39] (Fig 3). This variation in inflection points may be explained by mathematical methods used as well as other factors such as age, sex, race or ethnicity, calcium and phosphorus intake, weight status, extent of vitamin D insufficiency, pubertal status, or even inaccuracy of 25(OH)D assays [6, 40].

In the current study, we observed a serum 25(OH)D concentration of 20 ng/mL (95% CI: 7.1 to 32.2) was the point at which the iPTH concentration plateaued in adolescent girls, whereas no plateau was found in boys. In a study which was conducted on the 735 boys and girls aged 7–18 y with different ethnicities, the inflection point of 25(OH)D concentration for maximal suppression of PTH concentration was 37.0 (95% CI: 24.9 to 52.4) ng/mL. In subgroup analysis of sex, the value of inflection point for 25(OH)D was 23.0 (95% CI: -7.0 to 44.0) ng/mL in boys and 44.1 (95% CI: 31.2 to 54.9) ng/mL in girls [9]. However, consistent with our results, Hill TR et al in a study of Northern Ireland Young Heart's Project with 1015 Northern European Caucasian adolescents observed that a point of inflection (the serum 25(OH)D value at which PTH plateaus) for girls was 24 ng/ml; however, similar to our findings, among boys, they observed no plateau in PTH concentration when 25(OH)D concentration increased [6]. The reason for the difference between sexes is not apparent, but one possible reason may be related to sex hormone effects on skeletal metabolism within the sexes. The role of estrogen on bones mineralization is more prominent than androgens [41]. Furthermore, the differences in sunlight exposure, prevalence of obesity, physical activity levels, and dietary intakes between boys and girls might be other factors. It should be noted that among adults, PTH concentration response to vitamin D deficiency also differed between sexes [42]. We

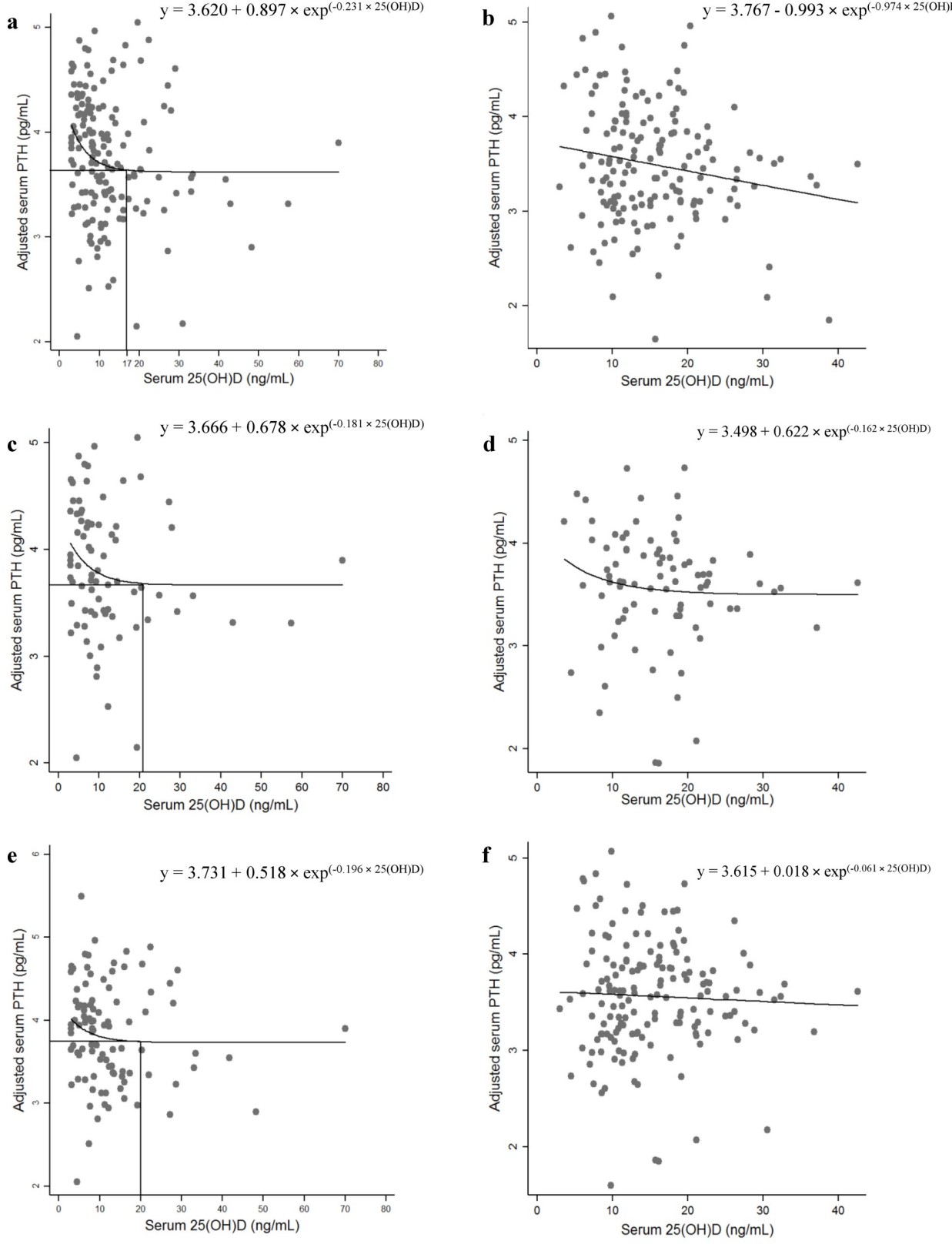

**Fig 2. Sensitivity analysis: to assess the association of serum 25(OH)D and adjusted log-transformed iPTH concentrations using nonlinear regression analysis.** (a) females with calcium intakes below EAR; (b) males with calcium intakes below EAR; (c) females with magnesium intakes below EAR; (d) males with magnesium intakes below EAR; (e) females with obesity; (f) males with obesity.

observed that girls had lower 25(OH)D concentrations and subsequently higher iPTH concentrations than that measured in boys, similar to the findings of Alyahya study [19]. Previous studies among adults also illustrate that women had significantly lower 25(OH)D concentrations and were more likely to be vitamin D deficient than males [43–45]. In this regard, the high proportion of girls (50%) under 10 ng/mL of 25(OH)D may lead to identify the point of inflection among girls. In our study, girls above and below the thresholds where iPTH increased had significantly different biochemical measurements including serum calcium, phosphorus, and alkaline phosphatase.

One explanation for variability in the point of inflection among studies may be related to the distribution of 25(OH)D concentration. It seems that, studies with higher threshold (>30 ng/mL) were attributed to higher 25(OH)D concentration (>20 ng/mL) [8, 9, 35, 37]. Although, in the current study, 84% of girls had serum 25(OH)D concentration lower than 20 ng/mL, which might influence the point of extracted threshold, there is also several studies in which high serum concentration of 25(OH)D were accompanied with low point of inflection [29, 34]. Therefore, a high serum 25(OH)D does not always lead to higher point of inflection in the 25(OH)D-PTH association. Also it should be kept in mind that findings of a study with

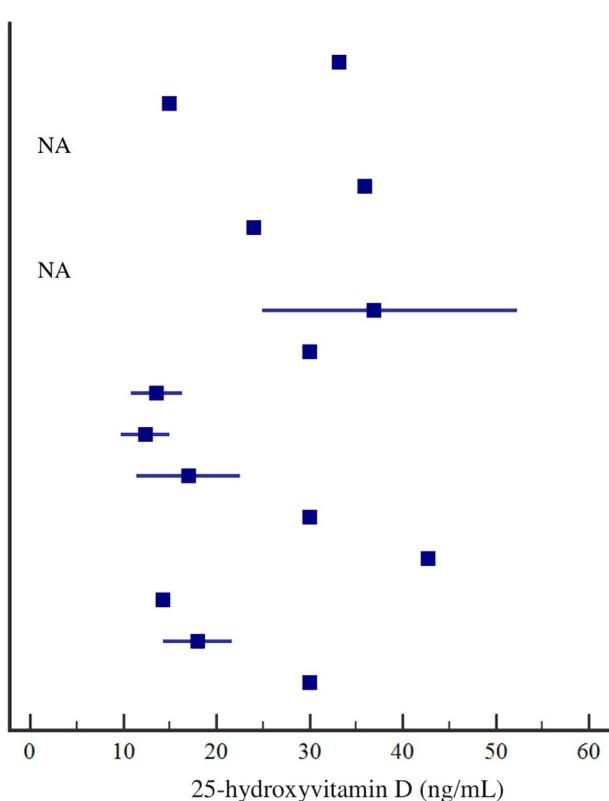

| Author | Sample | Numbers | 25-hydroxyvitamin D (ng/mL) |
|---|---|---|---|
| Guillemant J [37] | Boys | 175 | 23.4 (4.0)[3] |
| Lehtonen-Veromaa M [38] | Girls | 191 | 13.6 (5.5)[3] |
| Outila TA[30] | Girls | 178 | 15.6 (5.6)[3] |
| Harkness L[8] | Girls | 400 | 22.0 (12.0)[3] |
| Hill TR [6] | Girls | 510 | Not reported |
| Hill TR [6] | Boys | 505 | Not reported |
| Hill KM [9] | Both | 735 | 26.2 (10.2)[3] |
| Razzaghy-Azar M [32] | Both | 113 | Not reported |
| Atapattu N [39] | Both | 214 | 12.5 (0.2-66.5)[4] |
| Amini Z [29] | Both[1] | 93 | 25.5 (19.0-34.0)[4] |
| Amini Z [29] | Both[2] | 176 | 29.0 (22.5-40.0)[4] |
| Habibesadat S [33] | Both | 361 | 36.4 (15.5)[3] |
| Maguire JL[35] | Both | 1370 | 34.4 (11.2)[3] |
| Karaguzel G [36] | Both | 746 | 13.7 (7.3)[3] |
| Kang JI [34] | Both | 193 | 25.9 (3.07-79.8)[5] |
| Sahin ON [31] | Both | 3525 | Not reported |

**Fig 3. Thresholds defined in other studies.** [1] Overweight/obese, [2] Normal weight, [3] Mean (SD), [4] Median (IQR), [5] Mean (range).

a large number of children and adolescents from a broad range of latitudes and different races in the US indicated that the suppression point of PTH in relation to 25(OH)D concentration did not have difference among various races compared with pooled analyses [9].

Exploiting different statistical methods was another potential factor affecting extracted thresholds. Based on the aim of researchers, there were several statistical methods such as NLR, piecewise regression, and restricted cubic spline regression, which are frequently used to define a point in the reciprocal relationship between PTH and 25(OH)D concentration. However, in our study, when we determined the iPTH-25(OH)D nonlinear regression line, there were two main points in the iPTH-25(OH)D regression line which were important to be considered as thresholds (i.e., the point of plateau and the point of rapidly raised slope). By increasing the serum 25(OH)D concentration, iPTH concentration decreases smoothly while approaching the maximal suppression point. It is important to note that the slope of the line below the former threshold is almost 9 times greater than the slope of the line above this threshold. The critical issue is if the serum concentration of 25(OH) D is lower than a point of inflection (83.61% of girls < 20 ng/ml), it will release more iPTH (Fig 1), and it is arguable whether or not it could affect bone status. Given the importance of bone metabolism in iPTH-25(OH)D association, it may highlight the point at which the 25(OH)D concentration decreases and PTH concentration starts to rapidly rise. These inflection points may be used for defining deficiency or severe deficiency.

Since PTH concentration is predominantly regulated by calcium intake, as well as magnesium and phosphate intakes, their impacts on the association of serum iPTH-25(OH)D should be considered. Most of the studies defining a threshold for 25(OH)D concentration based on PTH concentration did not control for dietary intakes of these minerals. In order to overcome this deficit, in the current study, firstly dietary calcium intake was adjusted for iPTH concentrations; furthermore, in a sensitivity analysis, participants with calcium consumption>EAR were excluded. Our finding revealed that even after excluding individuals who consumed higher EAR dietary intakes of calcium, the point of maximal suppression of iPTH was not substantially changed.

Our study has some limitations. Since subgroup analysis on higher and lower consuming EAR of calcium and magnesium and obesity status was not pre-specified, we have no power to conduct analysis in all subgroups. Lack of data on calcium absorption is another limitation which should be mentioned. In addition, the analytical reliability of 25(OH)D assays was not monitored through the DEQAS (Vitamin D External Quality Assessment Scheme). Although data on race/ethnicity of the sample was not collected, current study consisted of a homogenous sample of Tehranian children and adolescents. This homogenous sample led to less generalizability of findings to other races. Further studies including different ethnic groups are warranted. Furthermore, The cross-sectional design limits the possibility of drawing definitive conclusions on the relationship between 25(OH)D and PTH. Prospective longitudinal study design should be conducted to verify the cross-sectional associations between 25(OH)D and PTH and associated changes in BMD and bone markers. Also, there might be other factors that could confound the association of PTH and 25(OH)D concentrations.

## Conclusion

This study revealed that in girls with excess weight, when concentrations of 25(OH)D increase above 20 ng/mL (95%CI: 7.1 to 32.2), an iPTH mean plateau level is reached, and when 25(OH)D concentrations approach 10 ng/mL (95%CI: 4.6 to 22.5), the slope in iPTH concentration accelerates.

## Acknowledgments

We would like to acknowledge Ms. Niloofar Shiva for a critical edition of English grammar and syntax and Dr Adriana Dusso for critical scientific comments on the manuscript.

## Author Contributions

**Conceptualization:** Emad Yuzbashian.

**Data curation:** Roya Shamsi.

**Formal analysis:** Maryam Mahdavi.

**Investigation:** Golaleh Asghari, Roya Shamsi.

**Methodology:** Golaleh Asghari, Emad Yuzbashian, Farhad Hosseinpanah.

**Supervision:** Farhad Hosseinpanah, Parvin Mirmiran.

**Writing – original draft:** Golaleh Asghari, Emad Yuzbashian, Carol L. Wagner, Maryam Mahdavi, Roya Shamsi, Farhad Hosseinpanah, Parvin Mirmiran.

**Writing – review & editing:** Golaleh Asghari, Emad Yuzbashian, Carol L. Wagner, Farhad Hosseinpanah, Parvin Mirmiran.

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
