## [Decision Letter · Decision Letter 0]

9 Jul 2019

PONE-D-19-15568

The relation between Circulating Levels of Vitamin D and Parathyroid Hormone in children and adolescent with overweight or obesity: Quest for a Threshold

PLOS ONE

Dear Dr Mirmiran,

Thank you for submitting your manuscript to PLOS ONE. After careful consideration, we feel that it has merit but does not fully meet PLOS ONE’s publication criteria as it currently stands. Therefore, we invite you to submit a revised version of the manuscript that addresses the points raised during the review process.

Many important points were identified by the reviewers for carefully revision of the manuscript. Please follow all recommendations, including English editing.  

We would appreciate receiving your revised manuscript by Aug 23 2019 11:59PM. To enhance the reproducibility of your results, we recommend that if applicable you deposit your laboratory protocols in protocols.io, where a protocol can be assigned its own identifier (DOI) such that it can be cited independently in the future. For instructions see: http://journals.plos.org/plosone/s/submission-guidelines#loc-laboratory-protocols

We look forward to receiving your revised manuscript.

Kind regards,

Marly A. Cardoso, Ph.D.

Academic Editor

PLOS ONE

Journal Requirements:

Please provide an amended Funding Statement that declares *all* the funding or sources of support received during this specific study (whether external or internal to your organization) as detailed online in our guide for authors at http://journals.plos.org/plosone/s/submit-now.  

Please state what role the funders took in the study.  If any authors received a salary from any of your funders, please state which authors and which funder. If the funders had no role, please state: "The funders had no role in study design, data collection and analysis, decision to publish, or preparation of the manuscript."

Reviewers' comments:

Reviewer's Responses to Questions

**Comments to the Author**

1. Is the manuscript technically sound, and do the data support the conclusions?

Reviewer #1: Yes

Reviewer #2: No

2. Has the statistical analysis been performed appropriately and rigorously? 

Reviewer #1: Yes

Reviewer #2: No

3. Have the authors made all data underlying the findings in their manuscript fully available?

Reviewer #1: No

Reviewer #2: No

4. Is the manuscript presented in an intelligible fashion and written in standard English?

Reviewer #1: Yes

Reviewer #2: No

5. Review Comments to the Author

Reviewer #1: This cross-sectional study aimed to identify 25(OH)D for maximal suppression of PTH in 378 boys and girls 6-14 years old in Tehran. This was a nicely done analysis, my major comments are regarding further elaboration on the literature on this topic and how this current study fits in. I believe this study has a huge strength in that the majority of the subjects have deficient 25OHD levels – this gives a unique opportunity to investigate the point of maximal PTH suppression in that very low range of 25OHD.

1) Please elaborate further on the findings of other studies of maximal suppression of PTH in children. Figure 3 includes many of these studies, but not all are even listed in the reference list. In Figure 3, please add the reference citation number, as well as the study sample size for each, as some studies are quite small and others larger. Harkness is misspelled in the Figure. I believe the “Hill (girls) 2010” is Hill TR 2010, and “Hill (both) 2010” is Hill KM 2010? the author first initials should be shown here for clarity. Additionally, Hill KM 2010 is one of the larger studies on this topic, included both boys and girls, and multiple race/ethnicities, so should be more prominently discussed throughout the paper.

2) What is the variation around the knot for maximal suppression in this study? Hill KM et al. show that even when the point estimate was defined, there was a large 95% confidence interval around this point. Most studies (as illustrated in Figure 3) do not describe the variation around the knot. Please give this variation for your results.

3) A much more extensive discussion of the differences between studies based on the distribution of 25OHD levels in the cohorts is needed. In the present study, it is stated that 84% of the girls were less than the 20 ng/mL threshold, perhaps making it more feasible to pinpoint a 25OHD level for maximal PTH suppression, in comparison to other studies (like Hill KM) that had a cohort with a distribution along much higher levels of 25OHD. The vitamin D status of the cohorts may also explain the variation around the knot for maximal suppression (i.e. greater in more adequate populations, tighter in more deficient populations).

4) Related to the comments above, this may be visually tricky – but consider whether figure 3 could also superimpose the median and distribution of the 25OHD levels with the point estimate and 95%CI(when available) for maximal PTH suppression. I think this could give huge insights into making sense of these various studies.

5) It is perhaps the greater 25OHD levels in boys that precludes the identification of a point of maximal suppression... the high proportion of girls under ~10 ng/mL likely drives the ability to identify the knot.

6) Hill TR is both Ref 7 and ref 37.

7) typo "accelarat" - last word of abstract

Reviewer #2: PONE-D-19-15568: The relation between Circulating Levels of Vitamin D and Parathyroid Hormone in children and adolescent with overweight or obesity: Quest for a Threshold

The present manuscript aims to investigate the association of circulating intact parathyroid hormone (iPTH) and 25(OH)D among overweight children and adolescents from primary schools in Tehran, with focus on 25(OH)D levels related to iPTH suppression. The manuscript, however, has major issues in properly approaching such questions. Also, a throughout English review is necessary.

In the introduction, explanation on maximal suppression and rapid raise in iPTH concentrations according to 25(OH)D levels should be more concise and clear. Fourth and fifth paragraphs are repetitive and fail to elucidate a role for overweight or obesity in such relationship. Authors must revise their rationale for conducting the present analysis.

In methods, recruitment of participants was not clearly explained. It also seems that data on race/ethnicity and sunlight exposure were not collected, which is problematic. Very importantly, authors did not present adequate justification or references for the generation of an adjusted sex-specific variable for iPTH --this is a crucial point since this is the variable used in all their analyses. In addition, it is not clear why three knots were used and why they were placed at the 25th, 50th and 75th centiles of 25(OH)D concentrations. Among the sensitivity analyses performed, it is not clear what authors meant by including obese children and adolescents in their analysis --have they been previously excluded? why?

Only after solving these points findings and discussion could be appraised. A better assessment of study limitations is indispensable, including study design, representativeness of the study population, and other potential confounders. Finally, please review your reference list --there are repeated citations (e.g., refs 4 & 29).

6. PLOS authors have the option to publish the peer review history of their article (what does this mean?). If published, this will include your full peer review and any attached files.

Reviewer #1: No

Reviewer #2: No

---

## [Author Response · Author response to Decision Letter 0]

26 Sep 2019

Response (R) to reviewer comments (C):

Reviewer # 1

C1) Please elaborate further on the findings of other studies of maximal suppression of PTH in children. Figure 3 includes many of these studies, but not all are even listed in the reference list. In Figure 3, please add the reference citation number, as well as the study sample size for each, as some studies are quite small and others larger. Harkness is misspelled in the Figure. I believe the “Hill (girls) 2010” is Hill TR 2010, and “Hill (both) 2010” is Hill KM 2010? the author first initials should be shown here for clarity. Additionally, Hill KM 2010 is one of the larger studies on this topic, included both boys and girls, and multiple race/ethnicities, so should be more prominently discussed throughout the paper.

R: We appreciate the reviewer for this valuable comment. We precisely revised figure 3 to address all deficits. Reference citation numbers and the study sample size of all included studies, as well as the study of Harkness L have been added to the Figure 3. Furthermore, we have discussed the findings of study which was conducted by Hill KM through the manuscripts as follows: 

Added on page 13, Lines 280-284: “In a study which was conducted on the 735 boys and girls aged 7–18 y with different ethnicities, the inflection point of 25(OH)D concentration for maximal suppression of PTH concentration was 37.0 (95% CI: 24.9 to 52.4) ng/mL. In subgroup analysis of gender, the value of inflection point for 25(OH)D was 23.0 (95% CI: -7.0 to 44.0) ng/mL in boys and 44.1 (95% CI: 31.2 to 54.9) ng/mL in girls [1].”

Added on page 14, Lines 309-312: “Also it should be kept in mind that findings of a study with a large number of children and adolescents from a broad range of latitudes and different races in the US indicated that the suppression point of PTH in relation to 25(OH)D concentration did not have difference among various races compared with pooled analyses [1].”

C2) What is the variation around the knot for maximal suppression in this study? Hill KM et al. show that even when the point estimate was defined, there was a large 95% confidence interval around this point. Most studies (as illustrated in Figure 3) do not describe the variation around the knot. Please give this variation for your results.

R: Thanks for your valuable comment. Piecewise linear regression modeling of log PTH for 25(OH)D showed that in the slope of iPTH-25(OH)D, iPTH began to rapidly rise at 10 ng/mL of 25(OH)D level among girls. Furthermore, finding from the formula extracted from non-linear association of 25(OH)D-PTH indicated that the point of plateau is around 20 ng/mL. Bootstrap resampling (n=5000) has been used to determine the 95% confidence interval around the point estimate of serum 25(OH)D for those points. The calculated 95% confidence intervals are 4.6 to 22.5 for the point of rapidly rise and 7.1 to 32.2 for point of plateau. The 95% confidence intervals have been added to the manuscripts as follows:

 Added on page 9, lines 178-180: “Bootstrap resampling (n=5000) has been used to determine the 95% confidence interval around the point estimate of serum 25(OH)D for point of plateau and rapidly rise of PTH”.

Added on page 10, lines 213-215: In girls (Fig 1, a): iPTH (pg/ ml) = 3.598 + 0.868 × exp(-0.190 × 25(OH)D) and a plateau in iPTH level at 44 pg/mL was observed at a serum 25(OH)D concentration of approximately 20 ng/mL (95% CI: 7.1 to 32.2). 

Added on page 11, lines 222-224: Piecewise linear regression modeling of log PTH for 25(OH)D showed that in the slope of iPTH-25(OH)D, iPTH began to rapidly rise at 10 ng/mL (95% CI: 4.6 to 22.5) of 25(OH)D level among girls (f: 9.8).

C3) A much more extensive discussion of the differences between studies based on the distribution of 25OHD levels in the cohorts is needed. In the present study, it is stated that 84% of the girls were less than the 20 ng/mL threshold, perhaps making it more feasible to pinpoint a 25OHD level for maximal PTH suppression, in comparison to other studies (like Hill KM) that had a cohort with a distribution along much higher levels of 25OHD. The vitamin D status of the cohorts may also explain the variation around the knot for maximal suppression (i.e. greater in more adequate populations, tighter in more deficient populations). 

R: Thanks to reviewer for providing this thoughtful comment. The distribution of 25(OH)D concentration might be another factor which can influence the points extracted from maximal PTH suppression. In this regard, we have added the central tendency (mean or median) and dispersion measures (SD, IQR, or range) as a column in the Figure 3. Distribution of 25(OH)D shows that studies with higher point of threshold (>30 ng/mL) reported higher 25(OH)D concentration (>20 ng/mL). However, lower suppression points (<30 ng/mL) are not always attributed to the low serum concentration of 25(OH)D. For example, Hill KM reported the mean of 26.2 (10.2) ng/mL and the threshold point of 37.0 ng/mL for 25(OH)D concentration, Kang et al and Amini et al with 25.9 and 25.5 mean of 25(OH)D concentration, respectively reported the point of threshold at 18.0 and 12.4, respectively. Therefore, a high serum 25(OH)D does not always lead to higher point of inflection in the 25(OH)D-PTH association. These finding have been discussed as follows:

Added on Pgae 14, lines 302-309: “One explanation for variability in the point of inflection among studies may be related to the distribution of 25(OH)D concentration. It seems that, studies with higher threshold (>30 ng/mL) were attributed to higher 25(OH)D concentration (>20 ng/mL) [1-4]. Although, in the current study, 84% of girls had serum 25(OH)D concentration lower than 20 ng/mL, which might influence the point of extracted threshold, there is also several studies in which high serum concentration of 25(OH)D were accompanied with low point of inflection [5, 6]. Therefore, a high serum 25(OH)D does not always lead to higher point of inflection in the 25(OH)D-PTH association.”

C4) Related to the comments above, this may be visually tricky – but consider whether figure 3 could also superimpose the median and distribution of the 25OHD levels with the point estimate and 95%CI(when available) for maximal PTH suppression. I think this could give huge insights into making sense of these various studies.

R: Agreed. Central tendency (mean or median) and dispersion measures (SD, IQR, or range) has been added to the Figure 3. The following statement has been added to the discussion: 

Added on Pgae 14, lines 302-309: “One explanation for variability in the point of inflection among studies may be related to the distribution of 25(OH)D concentration. It seems that, studies with higher threshold (>30 ng/mL) were attributed to higher 25(OH)D concentration (>20 ng/mL) [1-4]. Although, in the current study, 84% of girls had serum 25(OH)D concentration lower than 20 ng/mL, which might influence the point of extracted threshold, there is also several studies in which high serum concentration of 25(OH)D were accompanied with low point of inflection [5, 6]. Therefore, a high serum 25(OH)D does not always lead to higher point of inflection in the 25(OH)D-PTH association.”

C5) It is perhaps the greater 25OHD levels in boys that precludes the identification of a point of maximal suppression... the high proportion of girls under ~10 ng/mL likely drives the ability to identify the knot.

R: Agreed. The following explanation has been added to the discussion.

Added on page 14, lines 298 and 299: In this regard, the high proportion of girls (50%) under 10 ng/mL of 25(OH)D may lead to identify the point of inflection among girls. 

C6) Hill TR is both Ref 7 and ref 37.

R: Thanks for your consideration. The reference list has been reviewed and corrected.

C7) typo "accelarat" - last word of abstract

R: Agreed and corrected.  

Reviewer # 2

C1) In the introduction, explanation on maximal suppression and rapid raise in iPTH concentrations according to 25(OH)D levels should be more concise and clear. Fourth and fifth paragraphs are repetitive and fail to elucidate a role for overweight or obesity in such relationship. Authors must revise their rationale for conducting the present analysis.

R: We appreciate your thoughtful comment. The response to this comment has been provided in the following two parts:

1) Levels of serum 25(OH)D, considered to be the best indicator of overall vitamin D status [7, 8]. Knowledge regarding the 25(OH)D level at which serum PTH levels decrease and reach a plateau, also called the inflection point, is important in determining the cutoff point for vitamin D deficiency, and also for targeting vitamin D deficiency treatment [9]. In this regard, several studies were conducted to determine the inflection point for vitamin D based on non-linear association of 25(OH)D with PTH concentration [1, 3, 5, 6, 10, 11]. The non-linear curve of 25(OH)D-PTH-association also shows another important point. Therefore, two thresholds of 25(OH)D rather than a single inflection point is suggested. For example, in the following Figure by Kang et al [6], two important points in the 25(OH)D-PTH curve were suggested: 1) the 25(OH)D concentration is high enough to suppress the PTH concentration and 2) 25(OH)D has dropped enough to reciprocally increase PTH. 

Added on page 4, lines 72-80: “The inverse non-linear association between 25(OH)D concentration and parathyroid hormone (PTH) is considered to define the appropriate cut point for defining adequate vitamin D status in children and adults [12-14]. Studies indicated that there is two thresholds of 25(OH)D rather than a single inflection point in the 25(OH)D-PTH-association curve where: 1) the 25(OH)D concentration is high enough to suppress the PTH concentration and 2) 25(OH)D has dropped enough to reciprocally increase PTH. In fact, the former point is a threshold point for the 25(OH)D concentration at which serum PTH concentrations decrease and reach a plateau, and the latter point is a spot at which the intensity of PTH concentration in response to increasing 25(OH)D concentration dramatically changes and slowly reaching a maximal suppression point.”

2) The paragraphs four and five has been revised as the follows:

Added on page 5, lines 85-94: “Significant controversy exists regarding optimal vitamin D status in children and adolescents, which is complicated by certain factors such as excess weight. Alteration of the vitamin D endocrine system in obesity has been reported [15]. Excess body weight or fat accumulation in both adults and children are associated with lower 25(OH)D concentrations and higher PTH concentrations [16-19]. The 25(OH)D-PTH association may not be explained by the same mechanism in normal-weight individuals. It is not known whether the 25(OH)D-PTH association is impressed by obesity; however, there may be a different set-point for the 25(OH)D-PTH relationship in the obese pediatrics. Therefore, determining the threshold for 25(OH)D in children and adolescents with excess weight is more complex, and defining cut points in this population seems crucial.”

C2) In methods, recruitment of participants was not clearly explained. It also seems that data on race/ethnicity and sunlight exposure were not collected, which is problematic. Very importantly, authors did not present adequate justification or references for the generation of an adjusted sex-specific variable for iPTH --this is a crucial point since this is the variable used in all their analyses. 

R: Thanks for your comment. The response to this comment has been provided in the following three parts:

1) Participants were selected from primary schools located in three districts of Tehran (Iran). A list of primary schools in these districts was provided, and schools were randomly chosen in each district. Students with body mass index z-score> 1 (according to WHO criteria) were selected from the list of each school. The eligibility criteria were as follows: no known medical illnesses such as diabetes, kidney or liver disease (based on physician examination and medical records review), no taken any drug or supplementation, and no on specific diet during the past year. An alphabet list of all eligible students was prepared in Microsoft Excel, and then each student was marked with a specific number. Then a simple random sampling was generated by Microsoft Excel, and 378 students were identified. Finally, students and their guardians were invited to the Research Institute for Endocrine Sciences (RIES). It should be noted that simple random sampling applied in the current study means that each individual was chosen entirely by chance, and each member of the population has an equal chance of being included in the sample.

The sampling has been revised as follows:

Revised on page 5 and 6, lines 101-114: “The present cross-sectional study was conducted in Iran, Tehran located at 51̊ 24ˊE, 35̊ 42ˊ N from June 2016 to March 2017. Children and adolescents aged 6 to 13 years, with an age- and sex-specific body mass index (BMI) Z-scores ≥ 1 (according to criteria established by the World Health Organization), were recruited from primary schools located in three districts of Tehran. None of the adolescents had diabetes or other known medical illnesses such as liver or kidney diseases, associated with vitamin D metabolism (based on physician examination and medical records review), or used medication or supplements that might affect calciotropic hormones, or made intentional changes of dietary intake, or physical activity. An alphabet list of all eligible students was prepared and then a simple random sampling was generated. Finally, 180 girls and 198 boys meeting selection criteria were enrolled in the study. All children and adolescents and their guardians were invited to the Research Institute for Endocrine Sciences (RIES). The participants answered all questionnaires including socio-economic and health related issues, physical activity, and dietary intake. Height and weight were measured, and body mass index (BMI) was calculated. Stage of puberty was determined and a fasting blood sample was gathered.”

2) The sunlight exposure of participants was gathered using a questionnaire by asking about the daily duration of exposure to outdoor sunlight, the parts of body exposed to sunlight during this time, as well as the application of sunscreen. We have categorize participants based on their outdoor exposure to the sun; low sun exposure (<15 min/d) and high-sun-exposure (≥15 min/d). The data regarding sunlight exposure has been added to Table 1. Data on race/ethnicity had not been collected. It should be noted that we selected subjects from a homogenous population of Tehran. However, the lack of race/ethnicity information of population has been added to the limitation as follows: 

Added on page 7, lines 144-146: Exposure to sunlight was estimated using a questionnaire on daily duration of exposure to outdoor sunlight. Low sun-exposure was considered as exposing to sunlight<15min/d.

Added on page 16, lines 341-344: Although data on race/ethnicity of the sample was not collected, current study consisted of a homogenous sample of Tehranian children and adolescents. This homogenous sample led to less generalizability of findings to other races. Further studies including different ethnic groups are warranted.

3) There are several studies indicating that the concentration of PTH is gender-dependent biomarker [1, 10]. Available studies which investigated the association of 25(OH)D and PTH in children and adolescents showed that the concentration of PTH in boys and girls is significantly different [1, 10, 20, 21]. In our study, the mean PTH concentration was 39.8 pg/mL in boys and 60.2 pg/mL in girls. In addition, the mean and median adjusted PTH concentration when it has been generated in sex-specific manner is not equal to those has been generated on total sample. The importance of sex in the PTH concentration has been highlighted in the method section:

Added on page 8, lines 160-163: “an adjusted sex-specific variable (because of considerable difference between boys and girls in PTH concentration [21]) was generated for iPTH using its mean value plus the residuals obtained from regressing the iPTH level based on aforementioned confounding factors.”

C3) In addition, it is not clear why three knots were used and why they were placed at the 25th, 50th and 75th centiles of 25(OH)D concentrations. 

R: Thanks for your consideration. The statistical software uses basic functions for regression splines, based on knots, which positions are determined automatically according to equally spaced centiles of the distribution for continuous predictor. For example, by default three knots are placed at the 25th, 50th, and 75th centiles of any continuous predictor. When previous knowledge is available for the studied subject, the knots for a spline can be placed manually at values determined by such a knowledge. Therefore, when we know conditions are changed at particular value(s) of predictor, we can anticipate changes in the outcome [22].

C4) Among the sensitivity analyses performed, it is not clear what authors meant by including obese children and adolescents in their analysis --have they been previously excluded? why?

R: We appreciate the reviewer for this comment. In the current study we recruited children and adolescents with BMI Z-score ≥ 1. Therefore, our sample was consisted of children with overweight (BMI Z-score: 1 to 2) or obesity (BMI Z-score: ≥2). The main analysis has been performed on the boys and girls regardless of their weight status. In order to test the robustness of the results influenced by fat accumulation, we excluded subjects with overweight (BMI Z-score: 1 to 2) and conducted a sensitivity analysis limited to obese participants (BMI Z-score: ≥2). To prevent confusion, the sentence has been revised as follows:

Revised on page 9, lines 193-195: Excluding participants with overweight (BMI Z-score: 1 to 2), remaining participants with obesity (BMI Z-score ≥ 2) to account for the influence of obesity on serum iPTH concentrations.

C5) Only after solving these points findings and discussion could be appraised. A better assessment of study limitations is indispensable, including study design, representativeness of the study population, and other potential confounders. Finally, please review your reference list --there are repeated citations (e.g., refs 4 & 29).

R: The reference list has been reviewed and corrected. In addition, the following paragraph has been added to the limitation section:

Added on page 16, lines 341-349: Although data on race/ethnicity of the sample was not collected, current study consisted of a homogenous sample of Tehranian children and adolescents. This homogenous sample led to less generalizability of findings to other races. Further studies including different ethnic groups are warranted. Furthermore, The cross-sectional design limits the possibility of drawing definitive conclusions on the relationship between 25(OH)D and PTH. Prospective longitudinal study design should be conducted to verify the cross-sectional associations between 25(OH)D and PTH and associated changes in BMD and bone markers. Also, there might be other factors that could confound the association of PTH and 25(OH)D concentrations.

 

References: 

[1] K.M. Hill, G.P. McCabe, L.D. McCabe, C.M. Gordon, S.A. Abrams, C.M. Weaver, An inflection point of serum 25-hydroxyvitamin D for maximal suppression of parathyroid hormone is not evident from multi-site pooled data in children and adolescents, The Journal of nutrition, 140 (2010) 1983-1988.

[2] J. Guillemant, P. Taupin, H. Le, N. Taright, A. Allemandou, G. Peres, S. Guillemant, Vitamin D status during puberty in French healthy male adolescents, Osteoporosis International, 10 (1999) 222-225.

[3] L. Harkness, B. Cromer, Low levels of 25-hydroxy vitamin D are associated with elevated parathyroid hormone in healthy adolescent females, Osteoporosis International, 16 (2005) 109-113.

[4] J.L. Maguire, C. Birken, K.E. Thorpe, E.B. Sochett, P.C. Parkin, Parathyroid hormone as a functional indicator of vitamin D sufficiency in children, JAMA pediatrics, 168 (2014) 383-385.

[5] Z. Amini, S. Bryant, C. Smith, R. Singh, S. Kumar, Is the serum vitamin D-parathyroid hormone relationship influenced by obesity in children?, Horm Res Paediatr, 80 (2013) 252-256.

[6] J.I. Kang, Y.S. Lee, Y.J. Han, K.A. Kong, H.S. Kim, The serum level of 25-hydroxyvitamin D for maximal suppression of parathyroid hormone in children: the relationship between 25-hydroxyvitamin D and parathyroid hormone, Korean journal of pediatrics, 60 (2017) 45.

[7] M.F. Holick, Vitamin D deficiency, N Engl J Med, 357 (2007) 266-281.

[8] M.F. Holick, N.C. Binkley, H.A. Bischoff-Ferrari, C.M. Gordon, D.A. Hanley, R.P. Heaney, M.H. Murad, C.M. Weaver, Evaluation, treatment, and prevention of vitamin D deficiency: an Endocrine Society clinical practice guideline, J Clin Endocrinol Metab, 96 (2011) 1911-1930.

[9] B. Dawson-Hughes, R.P. Heaney, M.F. Holick, P. Lips, P.J. Meunier, R. Vieth, Estimates of optimal vitamin D status, Springer, 2005.

[10] T.R. Hill, A.A. Cotter, S. Mitchell, C.A. Boreham, W. Dubitzky, L. Murray, J.J. Strain, A. Flynn, P.J. Robson, J.M. Wallace, M. Kiely, K.D. Cashman, Vitamin D status and parathyroid hormone relationship in adolescents and its association with bone health parameters: analysis of the Northern Ireland Young Heart's Project, Osteoporosis international : a journal established as result of cooperation between the European Foundation for Osteoporosis and the National Osteoporosis Foundation of the USA, 21 (2010) 695-700.

[11] S. Habibesadat, K. Ali, J.M. Shabnam, A. Arash, Prevalence of vitamin D deficiency and its related factors in children and adolescents living in North Khorasan, Iran, Journal of Pediatric Endocrinology and Metabolism, 27 (2014) 431-436.

[12] P. Lips, Vitamin D deficiency and secondary hyperparathyroidism in the elderly: consequences for bone loss and fractures and therapeutic implications, Endocrine reviews, 22 (2001) 477-501.

[13] M.-C. Chapuy, P. Preziosi, M. Maamer, S. Arnaud, P. Galan, S. Hercberg, P. Meunier, Prevalence of vitamin D insufficiency in an adult normal population, Osteoporosis international, 7 (1997) 439-443.

[14] T. Hill, A. Cotter, S. Mitchell, C. Boreham, W. Dubitzky, L. Murray, J. Strain, A. Flynn, P. Robson, J. Wallace, Vitamin D Status and Parathyroid hormone relationship in adolescents and its association with bone health parameters: analysis of the Northern Ireland Young Heart’s Project, Osteoporosis international, 21 (2010) 695-700.

[15] Y. Liel, E. Ulmer, J. Shary, B.W. Hollis, N.H. Bell, Low circulating vitamin D in obesity, Calcif Tissue Int, 43 (1988) 199-201.

[16] C.B. Turer, H. Lin, G. Flores, Prevalence of vitamin D deficiency among overweight and obese US children, Pediatrics, (2012) peds. 2012-1711.

[17] J.L. Plesner, M. Dahl, C.E. Fonvig, T.R.H. Nielsen, J.T. Kloppenborg, O. Pedersen, T. Hansen, J.-C. Holm, Obesity is associated with vitamin D deficiency in Danish children and adolescents, Journal of Pediatric Endocrinology and Metabolism, 31 (2018) 53-61.

[18] S. Barja-Fernández, C.M. Aguilera, I. Martínez-Silva, R. Vazquez, M. Gil-Campos, J. Olza, J. Bedoya, C. Cadarso-Suárez, Á. Gil, L.M. Seoane, 25-Hydroxyvitamin D levels of children are inversely related to adiposity assessed by body mass index, Journal of physiology and biochemistry, 74 (2018) 111-118.

[19] L. Ke, R.S. Mason, L.A. Baur, C.T. Cowell, X. Liu, S.P. Garnett, K.E. Brock, Vitamin D levels in childhood and adolescence and cardiovascular risk factors in a cohort of healthy Australian children, The Journal of steroid biochemistry and molecular biology, 177 (2018) 270-277.

[20] M. Di Monaco, C. Castiglioni, F. Vallero, R. Di Monaco, R. Tappero, Parathyroid hormone response to severe vitamin D deficiency is sex associated: an observational study of 571 hip fracture inpatients, The journal of nutrition, health & aging, 17 (2013) 180-184.

[21] K.O. Alyahya, Vitamin D levels in schoolchildren: a cross-sectional study in Kuwait, BMC pediatrics, 17 (2017) 213-213.

[22] P. Royston, W. Sauerbrei, Multivariable modeling with cubic regression splines: a principled approach, The Stata Journal, 7 (2007) 45-70.

---

## [Decision Letter · Decision Letter 1]

29 Oct 2019

PONE-D-19-15568R1

The relation between Circulating Levels of Vitamin D and Parathyroid Hormone in children and adolescent with overweight or obesity: Quest for a Threshold

PLOS ONE

Dear Dr Mirmiran,

Thank you for submitting your manuscript to PLOS ONE. After careful consideration, we feel that it has merit but does not fully meet PLOS ONE’s publication criteria as it currently stands. Therefore, we invite you to submit a revised version of the manuscript that addresses the points raised during the review process.

The authors have made great improvements in the revised version of the manuscript. However, some points still need minor revision as suggested by the reviewer.

We would appreciate receiving your revised manuscript by Dec 13 2019 11:59PM. To enhance the reproducibility of your results, we recommend that if applicable you deposit your laboratory protocols in protocols.io, where a protocol can be assigned its own identifier (DOI) such that it can be cited independently in the future. For instructions see: http://journals.plos.org/plosone/s/submission-guidelines#loc-laboratory-protocols

We look forward to receiving your revised manuscript.

Kind regards,

Marly A. Cardoso, Ph.D.

Academic Editor

PLOS ONE

Reviewers' comments:

Reviewer's Responses to Questions

**Comments to the Author**

1. If the authors have adequately addressed your comments raised in a previous round of review and you feel that this manuscript is now acceptable for publication, you may indicate that here to bypass the “Comments to the Author” section, enter your conflict of interest statement in the “Confidential to Editor” section, and submit your "Accept" recommendation.

Reviewer #1: (No Response)

2. Is the manuscript technically sound, and do the data support the conclusions?

Reviewer #1: Yes

3. Has the statistical analysis been performed appropriately and rigorously? 

Reviewer #1: (No Response)

4. Have the authors made all data underlying the findings in their manuscript fully available?

Reviewer #1: Yes

5. Is the manuscript presented in an intelligible fashion and written in standard English?

Reviewer #1: Yes

6. Review Comments to the Author

Reviewer #1: Thank you for your responses to my previous comments. The addition of the bootstrap resampling method description and results is very helpful. I have a few further minor comments:

1) I appreciate the revisions to figure 3 to include reference numbers and sample sizes, etc. of each of the studies. There is a typographical error of "Hill MK" instead of "Hill KM".

2) Consider use of "sex" or "sexes" rather than "gender" or "genders" as "sex" is biological and "gender" is a social construct.

3) line 73 of new text: consider adding "commonly" ("is commonly considered to define")

4) line 91 of new text: suggest replacing "impressed" with "affected"

7. PLOS authors have the option to publish the peer review history of their article (what does this mean?). If published, this will include your full peer review and any attached files.

Reviewer #1: No

---

## [Author Response · Author response to Decision Letter 1]

30 Oct 2019

Editor-In-Chief, PLOS ONE

Greetings, 

Thank you very much for your email dated October 29th, 2019, regarding the secondary evaluation of our manuscript entitled “The relation between Circulating Levels of Vitamin D and Parathyroid Hormone in children and adolescent with overweight or obesity: Quest for a Threshold (PONE-D-19-15568R1)” and the opportunity to revise and resubmit the manuscript. We have complied with all reviewer comments, carefully considering each comment, and hope that the revised manuscript is now suitable for publication in Pols One. We submit the revised version, with a point by point response (R) to each reviewer comment (C) with changes highlighted. 

In the next page, we have addressed the data availability statement and financial disclosure.

Looking forward to hearing from you at your earliest convenience.

Sincerely yours,

Correspondence to: 

Parvin Mirmiran

Nutrition and Endocrine Research Center, Research Institute for Endocrine Science, Shahid Beheshti University of Medical Sciences, Tehran, Iran

P.O. Box: 19395-4763

Phone: +98 (21) 22432503

Fax: +98 (21) 22402463

E-mail: mirmiran@endocrine.ac.ir

Alternative E-mail: parvin.mirmiran@gmail.com

Co-correspondence to:

Farhad Hosseinpanah 

Obesity Research Center, Research Institute for Endocrine Sciences, Shahid Beheshti University of Medical Science, Tehran, Iran

P.O. Box: 19395-476 

Phone: +98-21-22432500 

Fax: +98-21-22416264

E-mail address: fhospanah@endocrine.ac.ir

1) “Availability of data” has been revised as the following: 

The data set is the property of Research Institute for Endocrine Sciences (RIES) and is made available upon approval of the research proposal by the research council and the ethics committee. The RIES ethics committee must issue an approval in case of a request for access to the de-identified dataset. Data request may be sent to the head of the RIES Ethics Committee Dr. Azita Zadeh-Vakili at email: azitavakili@endocrine.ac.ir.

2) In the manuscript, we revised the “Funding sources” section as the following:

This study was supported by the Shahid Beheshti University of Medical Sciences, Tehran, Iran (Grant no. 10429-4).The funders had no role in study design, data collection and analysis, decision to publish, or preparation of the manuscript. There was no additional external funding received for this study. 

 

Response (R) to reviewer comments (C):

Reviewer # 1

1) I appreciate the revisions to figure 3 to include reference numbers and sample sizes, etc. of each of the studies. There is a typographical error of "Hill MK" instead of "Hill KM".

2) Consider use of "sex" or "sexes" rather than "gender" or "genders" as "sex" is biological and "gender" is a social construct.

3) line 73 of new text: consider adding "commonly" ("is commonly considered to define")

4) line 91 of new text: suggest replacing "impressed" with "affected"

R: Thanks for your profound attention and comment on our manuscript. All above-mentioned concerns have been corrected through the text.

---

## [Editor Report · Decision Letter 2]

12 Nov 2019

The relation between Circulating Levels of Vitamin D and Parathyroid Hormone in children and adolescent with overweight or obesity: Quest for a Threshold

PONE-D-19-15568R2

Dear Dr. Mirmiran,

We are pleased to inform you that your manuscript has been judged scientifically suitable for publication and will be formally accepted for publication once it complies with all outstanding technical requirements.

With kind regards,

Marly A. Cardoso, Ph.D.

Academic Editor

PLOS ONE
---

## [Editor Report · Acceptance letter]

19 Nov 2019

PONE-D-19-15568R2 

The relation between Circulating Levels of Vitamin D and Parathyroid Hormone in children and adolescent with overweight or obesity: Quest for a Threshold 

Dear Dr. Mirmiran:

I am pleased to inform you that your manuscript has been deemed suitable for publication in PLOS ONE. Congratulations! Your manuscript is now with our production department. 

With kind regards,

on behalf of

Dr. Marly A. Cardoso 

Academic Editor

PLOS ONE